# Fixed-Time Aperiodic Intermittent Control for Quasi-Bipartite Synchronization of Competitive Neural Networks

**DOI:** 10.3390/e26030199

**Published:** 2024-02-26

**Authors:** Shimiao Tang, Jiarong Li, Haijun Jiang, Jinling Wang

**Affiliations:** 1College of Mathematics and System Science, Xinjiang University, Urumqi 830017, China; 2School of Mathematics and Statistics, Yili Normal University, Yining 835000, China; jianghaijunxju@163.com; 3College of Mathematics and Statistics, Northwest Normal University, Lanzhou 730070, China; jinling0803@126.com

**Keywords:** competitive neural network, quasi-bipartite synchronization, fixed-time intermittent control, external disturbance

## Abstract

This paper concerns a class of coupled competitive neural networks, subject to disturbance and discontinuous activation functions. To realize the fixed-time quasi-bipartite synchronization, an aperiodic intermittent controller is initially designed. Subsequently, by combining the fixed-time stability theory and nonsmooth analysis, several criteria are established to ensure the bipartite synchronization in fixed time. Moreover, synchronization error bounds and settling time estimates are provided. Finally, numerical simulations are presented to verify the main results.

## 1. Introduction

Since neural networks are widely used in computer science [1], remote sensing [2], autonomous control systems [3], and other fields, their dynamic behaviors have been extensively studied over the past several decades. It is worth noting that neural networks (NNs) consider the dynamic level of neural activity. However, it is essential to recognize that synaptic weights between neurons change over time [4]. Consequently, Meyer-bäse et al. [5] developed competitive neural networks (CNNs) with different time scales in 1996, which can be viewed as an extension of Hopfield neural networks and cellular networks [6,7]. CNNs are defined using two types of state variables: short-term memory (STM) describes rapid neural activity, while long-term memory (LTM) depicts slow, unsupervised synaptic modifications. On the other hand, coupled competitive neural networks (CCNNs) consist of several interconnected subsystems, and due to their complex dynamic behavior, they have garnered significant attention [8,9].

The activation functions of NNs are widely recognized for describing the connection between the input and output of a single neuron. They are commonly considered to be continuous. When the activation function is believed to be at the high gain limit, however, the activation function approaches discontinuity. As a result, an increasing number of scholars have been conducting considerable research on NNs with discontinuous activation functions [10,11,12,13]. On the other hand, the dynamic behaviors of NNs are frequently impacted by external disturbances, such as changes in network structure, hardware facilities, and environmental noise. As far as we are aware, there are few studies that take both discontinuous activation functions and external disturbances into account when discussing CCNNs. Therefore, it is both intriguing and challenging to research discontinuous activation functions and external disturbances in CCNNs.

Synchronization means that two or more dynamical systems achieve a common dynamical behavior. The synchronization problem of NNs has garnered significant attention recently due to its wide applicability in communication systems, biological sciences, mechanical engineering, and other domains [14,15,16,17]. However, the synchronization of the aforementioned NNs only considers the cooperative relationships between network nodes. In many practical systems, relationships of competition and cooperation coexist. Therefore, the synchronization issue of NNs with both competitive and cooperative connections between nodes, known as bipartite synchronization, is of crucial importance and has been researched in [18,19]. On the other hand, due to inherent network constraints, complete synchronization may not be achievable, and instead, quasi-synchronization is observed. Quasi-synchronization implies that the synchronization error no longer approaches zero but rather converges to a bounded set. To our knowledge, few papers have addressed the quasi-bipartite synchronization problem of CCNNs.

In addition to asymptotic synchronization or exponential synchronization, finite-time synchronization (FETs) has gained widespread attention as a more practical form of network synchronization in recent years. In FETs, the settling time is bounded, but it depends on the initial state of the node. To eliminate this dependence on the initial state, fixed-time synchronization (FXTs) is proposed based on the fixed-time (FXT) stability [20], and its settling time only depends on the system or control parameters. Compared with the rich results of NNs [21,22], it is extremely rare at present to explore the FXTs of CCNNs. In [23], the authors studied the finite-time bipartite synchronization of delayed CCNNs under quantized control. To our knowledge, there are limited reports on FXT quasi-bipartite synchronization of CCNNs, and further research is needed.

Over the past decades, the control problem of networks has been one of the most widely studied topics, and many useful control methods have been developed, such as adaptive control, sliding mode control, impulse control, and intermittent control, among others. Intermittent control involves alternating periods of applying control input and periods of no control input, making it a more economical choice compared to continuous control. Hence, the intermittent control strategy has received extensive attention [24,25,26,27]. In [28], the FXT synchronization problem of time-delay complex networks under intermittent pinning control is studied. The author in [29] solved the FXT and predefined-time cluster lag synchronization of stochastic multi-weighted complex networks via intermittent quantized control. To our knowledge, there is currently no existing literature that addresses the challenging problem of FXT quasi-bipartite synchronization of CCNNs under intermittent control.

Motivated by the analysis provided above, the primary objective of this paper is to investigate FXT quasi-bipartite synchronization in coupled competitive neural networks. Firstly, the model under consideration incorporates time-varying delays, discontinuous activation functions, and external disturbances simultaneously, rendering it more comprehensive. Secondly, we introduce an innovative FXT aperiodic intermittent control scheme, making the pioneering endeavor to explore quasi-bipartite synchronization in CCNNs. Furthermore, some robust criteria are established for FXT quasi-bipartite synchronization based on the theory of practical FXT stability. Finally, we provide estimations for error bounds and settling times.

The paper is organized as follows. Section 2 provides some necessary preliminary knowledge and the model description. Section 3 introduces the main theoretical conclusions. Section 4 provides numerical examples to validate the theoretical conclusions. Section 5 completes our study and discusses future research.

Notations: R represents the set of real numbers. The *n*-dimensional Euclidean space is represented by $R^{n}$. $R^{n\times m}$ is the set of n×m real matrices. $N_{+}=\left\{1,2,\cdots ,N\right\}$. $I_{n}$ is the identity matrix. $0_{n}$ denotes zero matrix. $\left|P\right|=\left(\right|p_{ij}{\left|\right)}_{n\times n}$. For a symmetric matrix *B*, $\lambda_{max}\left(B\right)$ represents the maximum eigenvalue of matrix *B*. diag(·) represents the diagonal matrix. $1_{n}$ denotes that all elements of a column vector are 1. For any $p={\left(p_{1},\cdots ,p_{n}\right)}^{T}\in R^{n}$, $\mathrm{sgn}\left(p\right)=\mathrm{diag}(\mathrm{sign}\left(p_{1}\right),\cdots ,\mathrm{sign}\left(p_{n}\right))$, sign(·) represents the sign function. $\mathrm{sign}\left(D\right)={\left(\mathrm{sign}\left(d_{ij}\right)\right)}_{n\times n}$. The 2-norm of the vector *p* is denoted by ${∥p∥}_{2}$. For vector w>0 (<,≥,≤), all of the components of w are positive (negative, non-negative, non-positive). For vectors $q_{1}$ and $q_{2}$, $q_{1}<q_{2}$ $\left(q_{1}\leq q_{2}\right)$ implies $q_{1}-q_{2}<0$ $\left(q_{1}-q_{2}\leq 0\right)$. Notation ⊗ denotes Kronecker product. k>0, $C\left(\left[-k,0\right],R^{n}\right)$ denotes the set of continuous function from [−k,0] to $R^{n}$.

## 2. Model Description and Preliminaries

Consider the following competitive neural networks with time-varying delay:(1)$$\left\{\begin{array}{c} \begin{array}{cc} \varepsilon {\dot{z}}_{k}\left(t\right)= & -c_{k}z_{k}\left(t\right)+\sum_{q=1}^{n}a_{kq}f_{q}\left(z_{q}\left(t\right)\right)+\sum_{q=1}^{n}b_{kq}f_{q}\left(z_{q}\left(t-\tau \left(t\right)\right)\right)\\ \; & +E_{k}\sum_{l=1}^{p}\omega_{l}m_{kl}\left(t\right),\;k=1,\cdots ,n,\\ {\dot{m}}_{kl}\left(t\right)= & -{\bar{c}}_{k}m_{kl}\left(t\right)+{\bar{a}}_{k}\omega_{l}f_{k}\left(z_{k}\left(t\right)\right),\;l=1,\cdots ,p, \end{array} \end{array}\right.$$
where $z_{k}\left(t\right)$ is the state variable. $m_{kl}\left(t\right)$ represents synaptic efficiency; $c_{k}>0$ represents the self-feedback coefficient. Connection weights and delay connection weights, respectively, are represented by $a_{kq}$ and $b_{kq}$. $f_{q}\left(\cdot \right)$ is the output of a neuron, which is discontinuous. $\omega_{l}$ is the weight of an external stimulus, ${\bar{c}}_{k}$ and ${\bar{a}}_{k}$ are given constants, $E_{k}$ is the intensity of the external stimulus, ε is the time scale of STM, and τ(t) is the time-varying delay.

**Remark** **1.**
*Neural network models are often formulated in terms of ordinary differential equations (ODEs), mainly because ODEs are a mathematical tool that effectively describes the dynamic behavior and interactions between neurons. Specifically: first, the state of the neuron needs to be defined. This may include the neuron’s potential, activity, or other relevant variables. These states will be the unknowns of the ODEs. Second, interaction rules need to be established. Rules describing the interactions between neurons are usually expressed in the form of weights and connections. These rules will determine how one neuron affects other neurons. Based on the states of neurons and the rules of interaction, a system of ODEs can be built to describe the dynamic evolution of a neural network. This usually involves differentiating the interaction rules to capture the temporal changes in the system. The ODE system then needs to be provided with initial conditions, which represent the initial state of the neural network at a given moment in time. Finally, the system of ODEs can be solved using numerical methods (e.g., Euler’s method, Runge–Kutta method, etc.) or analytical methods to obtain the state of the neural network at different points in time. In this way, the dynamic behavior of the neural network can be accurately represented in mathematical form. This model of ODEs not only helps in theoretical analysis, but also provides a basis for simulation and emulation, enabling us to better understand the behavior and response of neural networks.*


Let $s\left(t\right)={\left(s_{1}\left(t\right),\cdots ,s_{n}\left(t\right)\right)}^{T}$, $s_{k}\left(t\right)=\sum_{l=1}^{p}\omega_{l}m_{kl}\left(t\right)$, and $\omega ={\left(\omega_{1},\cdots ,\omega_{p}\right)}^{T}$. Without losing of generality, suppose ${∥\omega ∥}_{2}^{2}=1$, then network (1) can be written as
(2)$$\left\{\begin{array}{c} \begin{array}{cc} \dot{z}\left(t\right)= & -Cz(t)+Af(z(t))+Bf(z(t-\tau (t)))+Es(t),\\ \dot{s}\left(t\right)= & -\bar{C}s\left(t\right)+\bar{A}f\left(z\left(t\right)\right), \end{array} \end{array}\right.$$
where $z\left(t\right)={\left(z_{1}\left(t\right),\cdots ,z_{n}\left(t\right)\right)}^{T}$, $f\left(z\left(t\right)\right)={\left(f_{1}\left(z_{1}\left(t\right)\right),\cdots ,f_{n}\left(z_{n}\left(t\right)\right)\right)}^{T}$, $A=(\frac{a_{kq}}{\varepsilon }{)}_{n\times n}$, $B=(\frac{b_{kq}}{\varepsilon }{)}_{n\times n}$, $C=\mathrm{diag}(\frac{c_{1}}{\varepsilon },\cdots ,\frac{c_{n}}{\varepsilon })$, $\bar{C}=\mathrm{diag}\left({\bar{c}}_{1},\cdots ,{\bar{c}}_{n}\right)$, $\bar{A}=\mathrm{diag}\left({\bar{a}}_{1},\cdots ,{\bar{a}}_{n}\right)$, and $E=\mathrm{diag}(\frac{E_{1}}{\varepsilon },\cdots ,$ $\frac{E_{n}}{\varepsilon })$. The initial value of system (2) is given by $z\left(t\right)=\phi_{z}\left(t\right)\in C(\left[-\tau ,0\right],R^{n})$ and $s\left(t\right)=\phi_{s}\left(t\right)\in C(\left[-\tau ,0\right],R^{n})$.

A class of CCNNs with external disturbances are modeled as follows:(3)$$\left\{\begin{array}{c} \begin{array}{cc} {\dot{Z}}_{i}\left(t\right)= & -CZ_{i}\left(t\right)+Af\left(Z_{i}\left(t\right)\right)+Bf\left(Z_{i}\left(t-\tau \left(t\right)\right)\right)+ES_{i}\left(t\right)\\ \; & +\sum_{j=1}^{N}\left|d_{ij}\right|\left(\mathrm{sign}\left(d_{ij}\right)Z_{j}\left(t\right)-Z_{i}\left(t\right)\right)+R_{i}\left(t\right)+\Xi_{i}\left(t\right),\\ {\dot{S}}_{i}\left(t\right)= & -\bar{C}S_{i}\left(t\right)+\bar{A}f\left(Z_{i}\left(t\right)\right)+\sum_{j=1}^{N}\left|u_{ij}\right|\left(\mathrm{sign}\left(u_{ij}\right)S_{j}\left(t\right)-S_{i}\left(t\right)\right)\\ \; & +{\bar{R}}_{i}\left(t\right)+\Xi_{i}\left(t\right),i\in N_{+}, \end{array} \end{array}\right.$$
where $Z_{i}\left(t\right)={\left(Z_{i1}\left(t\right),\cdots ,Z_{in}\left(t\right)\right)}^{T}\in R^{n}$ and $S_{i}\left(t\right)={\left(S_{i1}\left(t\right),\cdots ,S_{in}\left(t\right)\right)}^{T}\in R^{n}$ are the state variables of STM and LTM, respectively; $\Xi_{i}\left(t\right)={\left(\Xi_{i1}\left(t\right),\cdots ,\Xi_{in}\left(t\right)\right)}^{T}$ indicates the external disturbance vector. $D=(d_{ij})$ and $U=\left(u_{ij}\right)\in R^{N\times N}$ are the adjacency matrix associated with the signed graph G(D) and G(U) of the CCNNs, satisfying $d_{ii}=0$ $\left(u_{ii}=0\right)$ for $i\in N_{+}$; for i≠j, $d_{ij}\neq 0$ $\left(u_{ij}\neq 0\right)$ if there is a directed communication link from node *j* to node *i*, otherwise $d_{ij}=0$ $\left(u_{ij}=0\right)$. $R_{i}\left(t\right)$ and ${\bar{R}}_{i}\left(t\right)$ are controllers to be designed. The initial conditions of network (3) meet: $Z_{i}\left(t\right)=\varphi_{i}\left(t\right)\in C(\left[-\tau ,0\right],R^{n})$ and $S_{i}\left(t\right)={\bar{\varphi }}_{i}\left(t\right)\in C(\left[-\tau ,0\right],R^{n})$.

**Remark** **2.**
*When $d_{ij}>0$ $\left(u_{ij}>0\right)$, then the connection between nodes i and j is cooperative, and the coupling term is given as $d_{ij}\left(Z_{j}\left(t\right)-Z_{i}\left(t\right)\right)$ $\left(u_{ij}\left(S_{j}\left(t\right)-S_{i}\left(t\right)\right)\right)$. When $d_{ij}<0$ $\left(u_{ij}<0\right)$, the connection between nodes i and j is competitive, and the coupling term is presented as $-d_{ij}\left(Z_{j}\left(t\right)+Z_{i}\left(t\right)\right)$ $\left(-u_{ij}\left(S_{j}\left(t\right)+S_{i}\left(t\right)\right)\right)$.*


For the convenience of discussion, define $x_{i}\left(t\right)=\left(Z_{i}^{T}\left(t\right),S_{i}^{T}\left(t\right)\right)$, $A_{1}={\left(A^{T},{\bar{A}}^{T}\right)}^{T}$, $B_{1}={\left(B^{T},0_{n}\right)}^{T}$, $W_{i}\left(t\right)={\left(\Xi_{i}^{T}\left(t\right),\Xi_{i}^{T}\left(t\right)\right)}^{T}$, $K_{i}\left(t\right)={\left(R_{i}^{T}\left(t\right),{\bar{R}}_{i}^{T}\left(t\right)\right)}^{T}$, $I=\left(I_{n},0_{n}\right)$, $C_{1}=\left[\begin{array}{cc} C & -E\\ 0_{n} & \bar{C} \end{array}\right]$, $d_{ij}=\left[\begin{array}{cc} d_{ij} & 0\\ 0 & u_{ij} \end{array}\right]⊗I_{n}$ and $D_{1}={\left(d_{ij}\right)}_{2nN\times 2nN}$.

Therefore, the coupled competitive neural networks (3) become:(4)$$\begin{array}{cc} {\dot{x}}_{i}\left(t\right)= & -C_{1}x_{i}\left(t\right)+A_{1}f\left(Ix_{i}\left(t\right)\right)+B_{1}f\left(Ix_{i}\left(t-\tau \left(t\right)\right)\right)+K_{i}\left(t\right)\\ \; & +\sum_{j=1}^{N}\left|d_{ij}\right|\left[\mathrm{Sign}\left(d_{ij}\right)x_{j}\left(t\right)-x_{i}\left(t\right)\right]+W_{i}\left(t\right). \end{array}$$

From (2), the tracking target can be described as follows:(5)$$\dot{y}\left(t\right)=-C_{1}y\left(t\right)+A_{1}f\left(Iy\left(t\right)\right)+B_{1}f\left(Iy\left(t-\tau \left(t\right)\right)\right),$$
where $y\left(t\right)={\left(z^{T}\left(t\right),s^{T}\left(t\right)\right)}^{T}$ is the state vector.

The necessary definitions, lemmas, and assumptions are given below.

**Definition** **1** ([30])**.** *Considering a system with discontinuous right-hand sides in the form of*
$$\dot{\zeta }\left(t\right)=F\left(\zeta \left(t\right)\right),\zeta \left(0\right)=0,$$
*where $\zeta \in R^{n}$, $F\left(\zeta \right):R^{n}\to R^{n}$ is locally bounded and Lebesgue measurable. The function ζ(t) is said to be the solution in the Filippov sense, which is defined in the interval $\left[0,t_{*}\right),\;t_{*}\in \left[0,+\infty \right)$, if ζ(t) is absolutely continuous and satisfies the below differential inclusion*
$$\dot{\zeta }\left(t\right)\in K\left[F\right]\left(\zeta \left(t\right)\right),a.e.t\in \left[0,t_{*}\right),$$
*where the set-valued map $K\left[F\right]:R^{n}\to R^{n}$ is defined as*
$$K\left[F\right]\left(\zeta \left(t\right)\right)≜\underset{\delta >0}{⋂}\underset{\mu (\Omega )>0}{⋂}\bar{co}\{F\left(B\left(\zeta \left(t\right),\delta \right)\setminus \Omega \right\},$$
*where $\bar{co}$ stands for the convex closure, μ(Ω) is the Lebesgue measure of the set *Ω*, and B(ζ(t),δ) denotes the open ball centered at ζ(t) with radius δ.*

**Definition** **2.**
*The network (4) is said to achieve FXT quasi-bipartite synchronization with network (5) if there is a constant $T_{1}>0$, such that*

(6)
$$\left\{\begin{array}{c} \lim_{t\to T_{1}}{∥x_{i}\left(t\right)-w_{i}y\left(t\right)∥}_{2}\leq \theta ,\\ ∥x_{i}\left(t\right)-w_{i}{y\left(t\right)∥}_{2}\leq \theta ,\;\forall t\geq T_{1},i\in N_{+}, \end{array}\right.$$

*where θ is a nonnegative constant.*


**Definition** **3** ([31])**.** *Aperiodically intermittent control is said to have an average control rate γ∈(0,1), if there is $T_{\gamma }\geq 0$ such that*
$$T_{con}\left(t,s\right)\geq \gamma \left(t-s\right)-T_{\gamma },\;\forall t>s>t_{0},$$
*where $T_{con}\left(t,s\right)$ denotes the total control interval length on [s,t), and $T_{\gamma }$ is called the elasticity number.*

**Lemma** **1** ([32])**.** *If $x_{i}\geq 0,i=1,\cdots ,n$, 0<ξ≤1 and η>1, then*
$$\sum_{i=1}^{n}x_{i}^{\xi }\geq {\left(\sum_{i=1}^{n}x_{i}\right)}^{\xi },\;\;\sum_{i=1}^{n}{x_{i}}^{\eta }\geq n^{1-\eta }{\left(\sum_{i=1}^{n}x_{i}\right)}^{\eta }.$$

**Lemma** **2** ([33])**.** *For any $x,y\in R^{n}$, and positive-definite matrix $\partial \in R^{n\times n}$, such that*
$$2x^{T}y\leq x^{T}\partial x+y^{T}\partial^{-1}y.$$

**Lemma** **3** ([34])**.** *Assume that there is a Lyapunov function V(t)≥0 that satisfies*
(7)$$\left\{\begin{array}{c} \dot{V}\left(t\right)\leq -a_{1}V^{q}\left(t\right)-a_{2}V^{p}\left(t\right)-a_{3}V\left(t\right)+{℘}_{1},\;t\in \left[\zeta_{k},\mu_{k}\right),\\ \dot{V}\left(t\right)\leq a_{4}V\left(t\right)+{℘}_{2},\;t\in \left[\mu_{k},\zeta_{k+1}\right), \end{array}\right.$$
*in which $a_{1},a_{2},a_{3},a_{4},{℘}_{1},{℘}_{2}$ are positive constants and 0<p<1, q>1. It is said the system practical fixed time is stable if $a_{4}-\gamma d<0$ and $\left(1-\gamma \right)d-a_{3}<0$, in which d>0. And the setting time $T^{*}$ satisfies*
$$T^{*}\leq \frac{1+a_{5}\left(q-1\right)T_{\gamma }}{a_{5}\left(q-1\right)\gamma }+\frac{1+a_{6}\left(1-p\right)T_{\gamma }}{a_{6}\left(1-p\right)\gamma },$$
*where $a_{5}=a_{1}\left(1-\phi \right)\exp\left\{T_{\gamma }\left(1-q\right)d\right\}$, $a_{6}=a_{2}\left(1-\phi \right)$ and 0<ϕ<1. γ and $T_{\gamma }$ are defined in Definition 3. When $t>T^{*}$, there is*
$$V\left(t\right)\leq \max\left\{\delta_{1},\delta_{2},\delta_{3}\right\},$$
*where $\delta_{1}={\left(\frac{{℘}_{1}}{\phi a_{1}}\right)}^{\frac{1}{q}}$, $\delta_{2}={\left(\frac{{℘}_{1}}{\phi a_{2}}\right)}^{\frac{1}{p}}$, $\delta_{3}=\frac{{℘}_{2}}{d\gamma -a_{4}}$.*

**Remark** **3.**
*When ${℘}_{1}$ and ${℘}_{2}$ are 0 in Lemma 3, it is still true, and we can refer to the proof of [35] Lemma 2. In this case, the result based on this lemma is complete synchronization. This lemma is more general and more widely applicable.*


For each q=1,⋯,n, the following assumptions are introduced:

**Assumption** **1** ([36])**.** *$f_{q}\left(\cdot \right):R\to R$ is continuous except on a countable set of isolated points $\left\{\theta_{r}^{q}\right\}$, where both the left limit $f_{q}^{-}\left(\theta_{r}^{q}\right)$ and right limit $f_{q}^{+}\left(\theta_{r}^{q}\right)$ exist. In addition, $f_{q}\left(\cdot \right)$ has at most finite discontinuous jump points in each bounded compact set.*

**Assumption** **2**([36])**.** *There exist positive constants $l_{q}$ and $h_{q}$ such that*
$$|\xi_{q}-\varsigma_{q}|\leq l_{q}\left|u-\nu \right|+h_{q},\forall u,\nu \in R,$$
*where $\xi_{q}\in K[f_{q}\left(u\right)]$, $\varsigma_{q}\in K[f_{q}\left(\nu \right)]$ with $K[f_{q}\left(\cdot \right)]=\bar{co}[f_{q}\left(\cdot \right)]=[min\left\{f_{q}^{-}\left(\cdot \right),f_{q}^{+}\left(\cdot \right)\right\},max\left\{f_{q}^{-}\left(\cdot \right),f_{q}^{+}\left(\cdot \right)\right\}]$.*

**Assumption** **3**([23])**.** *The signed graphs are structurally balanced. In other words, the node sets V of G can be divided into two unsigned subgraphs $V_{1}$ and $V_{2}$, respectively. It satisfies $V=V_{1}\cup V_{2}$ and $V_{1}\cap V_{2}=⌀$. In addition, the links inside each subgraph are nonnegative, while the links between two unsigned subgraphs are negative.*

**Assumption** **4.**
*The activation functions $f_{q}\left(\cdot \right)$ satisfies*

$$f_{q}\left(-z\right)=-f_{q}\left(z\right),\forall z\in R.$$



**Assumption** **5.**
*There exists a positive constant $M_{q}$ such that $|f_{q}\left(z\right)|\leq M_{q}$.*


**Assumption** **6.**
*The external disturbance $\Xi_{ik}\left(t\right)\left(k=1,\cdots ,2n\right)$ is bounded. That is, there is a positive constant $W_{k}$ such that $|\Xi_{ik}\left(t\right)|\leq W_{k}$.*


Assumption 3 implies that there exists a diagonal matrix $\omega =\mathrm{diag}(w_{1},\cdots ,w_{N})$, where $w_{i}=1$ if node $v_{i}\in V_{1}$, otherwise $w_{i}=-1$. To achieve the FXT quasi-bipartite synchronization of CCNNs, the intermittent controller is designed as follows:(8)$$R_{i}\left(t\right)=\left\{\begin{array}{c} -\lambda \left(x_{i}\left(t\right)-w_{i}y\left(t\right)\right)-\sigma_{1}\mathrm{sgn}\left(x_{i}\left(t\right)-w_{i}y\left(t\right)\right){\left|x_{i}\left(t\right)-w_{i}y\left(t\right)\right|}^{\alpha }\\ -\sigma_{2}\mathrm{sgn}\left(x_{i}\left(t\right)-w_{i}y\left(t\right)\right){\left|x_{i}\left(t\right)-w_{i}y\left(t\right)\right|}^{\beta },\;\;t\in \left[\zeta_{k},\mu_{k}\right),\\ 0_{2n}\;\;t\in \left[\mu_{k},\zeta_{k+1}\right), \end{array}\right.$$
where $i\in N_{+}$, 0<α<1, β>1, λ, $\sigma_{1}$, $\sigma_{2}$ are positive constants.

**Remark** **4.**
*The intermittent control proposed in this study can be accomplished in a fixed time, unlike the previous intermittent control, which can only achieve asymptotic results. Moreover, the aperiodic intermittent controller proposed in this study is different from the controller in [26,27] as follows:*

$$R_{i}\left(t\right)=\left\{\begin{array}{c} -\frac{r}{2}q\left(e_{i}\left(t\right)\right)-\frac{k}{2}\mathrm{sgn}\left(q\left(e_{i}\left(t\right)\right)\right)\left(|e_{i}\left(t\right){|}^{\alpha }+|e_{i}\left(t\right){|}^{\beta }\right)\\ -\frac{k}{2}({\left(\int_{t-\tau }^{t}e_{i}^{T}\left(s\right)e_{i}\left(s\right)ds\right)}^{\frac{1+\alpha }{2}}+(\int_{t-\tau }^{t}e_{i}^{T}\left(s\right)\\ \times e_{i}{\left(s\right)ds)}^{\frac{1+\beta }{2}})\frac{e_{i}\left(t\right)}{\|e_{i}{\left(t\right)\|}^{2}},\;\;\;\;\;\;\;\;\;\;\;\;\;\;\;\;\;\;\;\;\;\;\;\;\;mT\leq t<\left(m+\theta \right)T,\\ -\frac{r}{2}q\left(e_{i}\left(t\right)\right),\;\;\;\;\;\;\;\;\;\;\;\;\;\;\;\;\;\;\;\;\;\;\;\;\;\;\;\;\left(m+\theta \right)T\leq t<\left(m+1\right)T, \end{array}\right.$$

*and*

$$R_{i}\left(t\right)=\left\{\begin{array}{c} -k_{i}e_{i}\left(t\right)-\alpha \sum_{r=1}^{3}{\left(\frac{\xi_{r}}{1-\sigma_{r}}\int_{t-\tau_{r}\left(t\right)}^{t}e_{i}^{T}\left(s\right)e_{i}\left(s\right)ds\right)}^{\frac{p+1}{2}}\frac{e_{i}\left(t\right)}{\|e_{i}{\left(t\right)\|}^{2}}\\ -\beta \sum_{r=1}^{3}{\left(\frac{\xi_{r}}{1-\sigma_{r}}\int_{t-\tau_{r}\left(t\right)}^{t}e_{i}^{T}\left(s\right)e_{i}\left(s\right)ds\right)}^{\frac{q+1}{2}}\frac{e_{i}\left(t\right)}{\|e_{i}{\left(t\right)\|}^{2}}\\ -\alpha sign\left(e_{i}\left(t\right)\right)|e_{i}\left(t\right){|}^{p}-\beta sign\left(e_{i}\left(t\right)\right)|e_{i}\left(t\right){|}^{q},\;\;mT\leq t<\left(m+\theta \right)T,\\ -k_{i}e_{i}\left(t\right),\;\;\;\;\;\;\;\;\;\;\;\;\;\;\;\left(m+\theta \right)T\leq t<\left(m+1\right)T. \end{array}\right.$$

*A linear term does not need to be set in the rest interval. This approach is proposed first to achieve FXT quasi-bipartite synchronization for coupled competitive neural networks. Furthermore, aperiodic intermittent control can be degenerated into periodic intermittent control and continuous control. It is particularly suitable for complex systems that require dynamic and flexible control.*


Combined with controller (8), when $t\in [\zeta_{k},\mu_{k})$, the CCNNs (4) is rewritten as follows:(9)$$\begin{array}{cc} {\dot{x}}_{i}\left(t\right)= & -C_{1}x_{i}\left(t\right)+A_{1}f\left(Ix_{i}\left(t\right)\right)+B_{1}f\left(Ix_{i}\left(t-\tau \left(t\right)\right)\right)+\sum_{j=1}^{N}{\bar{d}}_{ij}x_{j}\left(t\right)+W_{i}\left(t\right)\\ \; & -\lambda \left(x_{i}\left(t\right)-w_{i}y\left(t\right)\right)-\sigma_{1}\mathrm{sgn}\left(x_{i}\left(t\right)-w_{i}y\left(t\right)\right){\left|x_{i}\left(t\right)-w_{i}y\left(t\right)\right|}^{\alpha }\\ \; & -\sigma_{2}\mathrm{sgn}\left(x_{i}\left(t\right)-w_{i}y\left(t\right)\right){\left|x_{i}\left(t\right)-w_{i}y\left(t\right)\right|}^{\beta }, \end{array}$$
where ${\bar{D}}_{1}={\left({\bar{d}}_{ij}\right)}_{2nN\times 2nN}$, ${\bar{d}}_{ij}=d_{ij}$ for i≠j and ${\bar{d}}_{ii}=-\sum_{j=1,j\neq i}^{N}\left|d_{ij}\right|$. Then, according to Assumption 3, the CCNNs (9) can be transformed into:(10)$$\begin{array}{cc} {\dot{\tilde{x}}}_{i}\left(t\right)= & -C_{1}{\tilde{x}}_{i}\left(t\right)+A_{1}f\left(I{\tilde{x}}_{i}\left(t\right)\right)+B_{1}f\left(I{\tilde{x}}_{i}\left(t-\tau \left(t\right)\right)\right)+\sum_{j=1}^{N}{\tilde{d}}_{ij}{\tilde{x}}_{j}\left(t\right)\\ \; & -\lambda \left({\tilde{x}}_{i}\left(t\right)-y\left(t\right)\right)-\sigma_{1}\mathrm{sgn}\left({\tilde{x}}_{i}\left(t\right)-y\left(t\right)\right){\left|{\tilde{x}}_{i}\left(t\right)-y\left(t\right)\right|}^{\alpha }\\ \; & -\sigma_{2}\mathrm{sgn}\left({\tilde{x}}_{i}\left(t\right)-y\left(t\right)\right){\left|{\tilde{x}}_{i}\left(t\right)-y\left(t\right)\right|}^{\beta }+w_{i}W_{i}\left(t\right), \end{array}$$
where ${\tilde{x}}_{i}\left(t\right)=w_{i}x_{i}\left(t\right)$, ${\tilde{d}}_{ij}=w_{i}{\bar{d}}_{ij}w_{j}=\left|d_{ij}\right|$ for i≠j and ${\tilde{d}}_{ii}=-\sum_{j=1,j\neq i}^{N}\left|d_{ij}\right|$, $i\in N_{+}$.

In a similar analysis, for $t\in [\mu_{k},\zeta_{k+1})$, one has
(11)$$\begin{array}{cc} {\dot{\tilde{x}}}_{i}\left(t\right)= & -C_{1}{\tilde{x}}_{i}\left(t\right)+A_{1}f\left(I{\tilde{x}}_{i}\left(t\right)\right)+B_{1}f\left(I{\tilde{x}}_{i}\left(t-\tau \left(t\right)\right)\right)\\ \; & +\sum_{j=1}^{N}{\tilde{d}}_{ij}{\tilde{x}}_{j}\left(t\right)+w_{i}W_{i}\left(t\right). \end{array}$$

By Definition 1, there is $\varpi \left(t\right)={\left(\varpi_{1}\left(t\right),\cdots ,\varpi_{n}\left(t\right)\right)}^{T}\in K[f(Iy)]$ such that
(12)$$\dot{y}\left(t\right)=-C_{1}y\left(t\right)+A_{1}\varpi \left(t\right)+B_{1}\varpi \left(t-\tau \left(t\right)\right).$$

Similarly, there exists at least one measurable function $\hbar_{i}\left(t\right)=(\hbar_{i1}\left(t\right),\cdots ,$ $\hbar_{in}\left(t\right){)}^{T}\in K[f\left(I{\tilde{x}}_{i}\right)]$ such that
(13)$$\left\{\begin{array}{c} \begin{array}{cc} {\dot{\tilde{x}}}_{i}\left(t\right)= & -C_{1}{\tilde{x}}_{i}\left(t\right)+A_{1}\hbar_{i}\left(t\right)+B_{1}\hbar_{i}\left(t-\tau \left(t\right)\right)+\sum_{j=1}^{N}{\tilde{d}}_{ij}{\tilde{x}}_{j}\left(t\right)+w_{i}W_{i}\left(t\right)\\ \; & -\lambda \left({\tilde{x}}_{i}\left(t\right)-y\left(t\right)\right)-\sigma_{1}\mathrm{sgn}\left({\tilde{x}}_{i}\left(t\right)-y\left(t\right)\right){\left|{\tilde{x}}_{i}\left(t\right)-y\left(t\right)\right|}^{\alpha }\\ \; & -\sigma_{2}\mathrm{sgn}\left({\tilde{x}}_{i}\left(t\right)-y\left(t\right)\right){\left|{\tilde{x}}_{i}\left(t\right)-y\left(t\right)\right|}^{\beta },\;t\in \left[\zeta_{k},\mu_{k}\right),\\ {\dot{\tilde{x}}}_{i}\left(t\right)= & -C_{1}{\tilde{x}}_{i}\left(t\right)+A_{1}\hbar_{i}\left(t\right)+B_{1}\hbar_{i}\left(t-\tau \left(t\right)\right)\\ \; & +\sum_{j=1}^{N}{\tilde{d}}_{ij}{\tilde{x}}_{j}\left(t\right)+w_{i}W_{i}\left(t\right),\;t\in \left[\mu_{k},\zeta_{k+1}\right). \end{array} \end{array}\right.$$

## 3. Main Result

In this part, we mainly study the fixed-time quasi-bipartite synchronization of networks (12) and (13) with external disturbances. According to the fixed-time stability theory, under the designed controller (8), by constructing an appropriate Lyapunov function, some criteria are derived to ensure that networks (12) and (13) achieve fixed-time quasi-bipartite synchronization.

Define the synchronization error $e_{i}\left(t\right)={\tilde{x}}_{i}\left(t\right)-y\left(t\right)$, and then the error dynamical system is:(14)$$\left\{\begin{array}{c} \begin{array}{cc} {\dot{e}}_{i}\left(t\right)= & -C_{1}e_{i}\left(t\right)+A_{1}g_{i}\left(t\right)+B_{1}g_{i}\left(t-\tau \left(t\right)\right)+\sum_{j=1}^{N}{\tilde{d}}_{ij}e_{j}\left(t\right)+w_{i}W_{i}\left(t\right)\\ \; & -\sigma_{1}\mathrm{sgn}\left(e_{i}\left(t\right)\right)|e_{i}{\left(t\right)|}^{\alpha }-\sigma_{2}\mathrm{sgn}\left(e_{i}\left(t\right)\right){\left|e_{i}\left(t\right)\right|}^{\beta }\\ \; & -\lambda \left(e_{i}\left(t\right)\right),\;t\in \left[\zeta_{k},\mu_{k}\right),\\ {\dot{e}}_{i}\left(t\right)= & -C_{1}e_{i}\left(t\right)+A_{1}g_{i}\left(t\right)+B_{1}g_{i}\left(t-\tau \left(t\right)\right)+\sum_{j=1}^{N}{\tilde{d}}_{ij}e_{j}\left(t\right)\\ \; & +w_{i}W_{i}\left(t\right),\;t\in \left[\mu_{k},\zeta_{k+1}\right), \end{array} \end{array}\right.$$
where $g_{i}\left(t\right)=\hbar_{i}\left(t\right)-\varpi \left(t\right)$, $g_{i}\left(t-\tau \left(t\right)\right)=\hbar_{i}\left(t-\tau \left(t\right)\right)-\varpi \left(t-\tau \left(t\right)\right)$.

**Theorem** **1.**
*Based on Assumptions 1–6 and the controller (8), if*

(15)
$$\begin{array}{cc} \; & 2\lambda -\lambda_{\max}\left(\Phi \right)-\kappa >0,\\ \; & \psi +\lambda_{\max}\left(\Phi \right)+\kappa >0,\\ \; & \psi +\lambda_{\max}\left(\Phi \right)+\kappa -\gamma d<0,\\ \; & \left(1-\gamma \right)d-2\lambda +\lambda_{\max}\left(\Phi \right)+\kappa <0, \end{array}$$

*where $\Phi =-I_{N}⊗\left(C_{1}+C_{1}^{T}\right)+I_{N}⊗\left(|A_{1}\left|LI+(\right|A_{1}{\left|LI\right)}^{T}\right)+\left({\tilde{D}}_{1}+{\tilde{D}}_{1}^{T}\right)$, $\kappa =\kappa_{1}+2\kappa_{2}+\kappa_{3}$, ψ, λ, d are positive constants, $\gamma \in \left(0,1\right)$ stands for the average control rate, then the quasi-bipartite synchronization can be ensured between networks (12) and (13) in fixed time. The settling time satisfies*

$$T^{*}\leq \frac{2+a_{5}\left(\beta -1\right)T_{\gamma }}{a_{5}\left(\beta -1\right)\gamma }+\frac{2+a_{6}\left(1-\alpha \right)T_{\gamma }}{a_{6}\left(1-\alpha \right)\gamma },$$

*Here, $a_{5}=2\sigma_{2}{\left(2nN\right)}^{\frac{1-\beta }{2}}\left(1-\phi \right)\exp\left\{\frac{1}{2}T_{\gamma }\left(1-\beta \right)d\right\}$, $a_{6}=2\sigma_{1}\left(1-\phi \right)$, $T_{\gamma }$ represents the elasticity number, 0<ϕ<1. The state trajectory of (14) converges to a compact set $\Omega =\left\{e\left(t\right)|{∥e\left(t\right)∥}_{2}\leq \sqrt{\max\{\delta_{1},\delta_{2},\delta_{3}\}}\right\}$ within $T^{*}$, $\delta_{1}={\left(\frac{ℵ}{2\phi \sigma_{2}{\left(nN\right)}^{\frac{1-\beta }{2}}}\right)}^{\frac{2}{\beta +1}}$, $\delta_{2}={\left(\frac{ℵ}{2\phi \sigma_{1}}\right)}^{\frac{2}{\alpha +1}}$, $\delta_{3}=\frac{ℵ}{d\gamma -\tilde{\psi }}$, $\tilde{\psi }=\psi +\lambda_{\max}\left(\Phi \right)+\kappa$, $ℵ=\frac{N}{\kappa_{1}}{\left(|A_{1}|h\right)}^{T}|A_{1}|h+\frac{2N}{\kappa_{2}}{\left(|B_{1}|M\right)}^{T}\left|B_{1}\right|M+\frac{N}{\kappa_{3}}W^{T}W$.*


**Proof.** Consider the following Lyapunov function
$$V\left(t\right)=e^{T}\left(t\right)e\left(t\right),$$
where $e\left(t\right)={\left(e_{1}^{T}\left(t\right),\cdots ,e_{N}^{T}\left(t\right)\right)}^{T}$.For $t\in \left[\zeta_{k},\mu_{k}\right)$, calculate the derivative of V(t) along the trajectory of the error system (14), and one has
$$\begin{array}{cc} \dot{V}\left(t\right)= & 2\sum_{i=1}^{N}e_{i}^{T}\left(t\right)(-C_{1}e_{i}\left(t\right)+A_{1}g_{i}\left(t\right)+B_{1}g_{i}\left(t-\tau \left(t\right)\right)+\sum_{j=1}^{N}{\tilde{d}}_{ij}e_{j}\left(t\right)\\ \; & +w_{i}W_{i}\left(t\right)-\lambda \left(e_{i}\left(t\right)\right)-\sigma_{1}\mathrm{sgn}\left(e_{i}\left(t\right)\right)|e_{i}{\left(t\right)|}^{\alpha }-\sigma_{2}\mathrm{sgn}\left(e_{i}\left(t\right)\right){\left|e_{i}\left(t\right)\right|}^{\beta }). \end{array}$$Based on Assumption 2, we obtain
$$\begin{array}{cc} 2\sum_{i=1}^{N}e_{i}^{T}\left(t\right)A_{1}g_{i}\left(t\right) & \leq 2\sum_{i=1}^{N}|e_{i}^{T}\left(t\right)||A_{1}\left|\right|\hbar_{i}\left(t\right)-\varpi \left(t\right)|\\ \; & \leq 2\sum_{i=1}^{N}|e_{i}^{T}\left(t\right)||A_{1}\left|[LI\right|e_{i}\left(t\right)|+h], \end{array}$$
where $L=\mathrm{diag}\{l_{1},\cdots ,l_{n}\}$, $h={\left(h_{1},\cdots ,h_{n}\right)}^{T}$.By Assumption 5, there is
(16)$$\begin{array}{cc} 2\sum_{i=1}^{N}e_{i}^{T}\left(t\right)B_{1}g_{i}\left(t-\tau \left(t\right)\right) & \leq 2\sum_{i=1}^{N}|e_{i}^{T}\left(t\right)||B_{1}\left|\right|\hbar_{i}\left(t-\tau \left(t\right)\right)-\varpi \left(t-\tau \left(t\right)\right)|\\ \; & \leq 4\sum_{i=1}^{N}|e_{i}^{T}\left(t\right)||B_{1}\left|\right|M, \end{array}$$
where $M={\left(M_{1},\cdots ,M_{n}\right)}^{T}$.From (15) and (16), one has
$$\begin{array}{cc} \dot{V}\left(t\right)\leq & -2\sum_{i=1}^{N}e_{i}^{T}\left(t\right)C_{1}e_{i}\left(t\right)+2\sum_{i=1}^{N}|e_{i}^{T}\left(t\right)||A_{1}\left|LI\right|e_{i}\left(t\right)|+2\sum_{i=1}^{N}|e_{i}^{T}\left(t\right)||A_{1}|h\\ \; & +2\sum_{i=1}^{N}e_{i}^{T}\left(t\right)\sum_{j=1}^{N}{\tilde{d}}_{ij}e_{j}\left(t\right)+2\sum_{i=1}^{N}e_{i}^{T}\left(t\right)|W_{i}\left(t\right)|-2\lambda \sum_{i=1}^{N}e_{i}^{T}\left(t\right)e_{i}\left(t\right)\\ \; & -2\sigma_{1}\sum_{i=1}^{N}e_{i}^{T}\left(t\right)\mathrm{sgn}\left(e_{i}\left(t\right)\right)|e_{i}{\left(t\right)|}^{\alpha }-2\sigma_{2}\sum_{i=1}^{N}e_{i}^{T}\left(t\right)\mathrm{sgn}\left(e_{i}\left(t\right)\right){\left|e_{i}\left(t\right)\right|}^{\beta }\\ \; & +4\sum_{i=1}^{N}|e_{i}^{T}\left(t\right)||B_{1}|M. \end{array}$$According to Lemma 1, one has
(17)$$\begin{array}{cc} -2\sigma_{1}\sum_{i=1}^{N}e_{i}^{T}\left(t\right)\mathrm{sgn}\left(e_{i}\left(t\right)\right){\left|e_{i}\left(t\right)\right|}^{\alpha } & =-2\sigma_{1}\sum_{i=1}^{N}\sum_{k=1}^{2n}{\left|e_{ik}\left(t\right)\right|}^{\alpha +1}\\ \; & \leq -2\sigma_{1}\sum_{i=1}^{N}{\left(\sum_{k=1}^{2n}e_{ik}^{2}\left(t\right)\right)}^{\frac{\alpha +1}{2}}\\ \; & \leq -2\sigma_{1}{\left(\sum_{i=1}^{N}e_{i}^{T}\left(t\right)e_{i}\left(t\right)\right)}^{\frac{\alpha +1}{2}}, \end{array}$$
and
(18)$$\begin{array}{cc} -2\sigma_{2}\sum_{i=1}^{N}e_{i}^{T}\left(t\right)\mathrm{sgn}\left(e_{i}\left(t\right)\right){\left|e_{i}\left(t\right)\right|}^{\beta } & =-2\sigma_{2}\sum_{i=1}^{N}\sum_{k=1}^{2n}{\left|e_{ik}\left(t\right)\right|}^{\beta +1}\\ \; & \leq -2\sigma_{2}\sum_{i=1}^{N}\sum_{k=1}^{2n}(e_{ik}^{2}\left(t\right){)}^{\frac{\beta +1}{2}}\\ \; & \leq -2{\left(2nN\right)}^{\frac{1-\beta }{2}}\sigma_{2}{\left(\sum_{i=1}^{N}e_{i}^{T}\left(t\right)e_{i}\left(t\right)\right)}^{\frac{\beta +1}{2}}. \end{array}$$From Lemma 2, it follows that
(19)$$\begin{array}{c} 2\sum_{i=1}^{N}|e_{i}^{T}\left(t\right)||A_{1}|h\leq \kappa_{1}\sum_{i=1}^{N}|e_{i}^{T}\left(t\right)||e_{i}\left(t\right)|+\frac{N}{\kappa_{1}}\left(\right|A_{1}|h{)}^{T}\left|A_{1}\right|h, \end{array}$$
(20)$$\begin{array}{c} 4\sum_{i=1}^{N}|e_{i}^{T}\left(t\right)||B_{1}|M\leq 2\kappa_{2}\sum_{i=1}^{N}|e_{i}^{T}\left(t\right)||e_{i}\left(t\right)|+\frac{2N}{\kappa_{2}}\left(\right|B_{1}|M{)}^{T}\left|B_{1}\right|M, \end{array}$$
and
(21)$$\begin{array}{cc} 2\sum_{i=1}^{N}e_{i}^{T}\left(t\right)|W_{i}\left(t\right)|\leq & \kappa_{3}\sum_{i=1}^{N}e_{i}^{T}\left(t\right)e_{i}\left(t\right)+\frac{N}{\kappa_{3}}W^{T}W, \end{array}$$
where $W={\left(W_{1},\cdots ,W_{2n}\right)}^{T}.$It follows from (17)–(21) that
$$\begin{array}{l} \begin{array}{cc} \dot{V}\left(t\right)\leq & -2\sum_{i=1}^{N}e_{i}^{T}\left(t\right)C_{1}e_{i}\left(t\right)+2\sum_{i=1}^{N}|e_{i}^{T}\left(t\right)||A_{1}\left|LI\right|e_{i}\left(t\right)|+\kappa_{1}\sum_{i=1}^{N}|e_{i}^{T}\left(t\right)||e_{i}\left(t\right)|\\ \; & +2\sum_{i=1}^{N}\sum_{j=1}^{N}e_{i}^{T}\left(t\right){\tilde{d}}_{ij}e_{j}\left(t\right)+2\kappa_{2}\sum_{i=1}^{N}|e_{i}^{T}\left(t\right)||e_{i}\left(t\right)|+\frac{N}{\kappa_{1}}\left(\right|A_{1}|h{)}^{T}\left|A_{1}\right|h\\ \; & +\frac{2N}{\kappa_{2}}\left(\right|B_{1}|M{)}^{T}\left|B_{1}\right|M-2\lambda \sum_{i=1}^{N}e_{i}^{T}\left(t\right)e_{i}\left(t\right)-2\sigma_{1}(\sum_{i=1}^{N}e_{i}^{T}\left(t\right)e_{i}\left(t\right){)}^{\frac{\alpha +1}{2}}\\ \; & -2\sigma_{2}(2nN{)}^{\frac{1-\beta }{2}}(\sum_{i=1}^{N}e_{i}^{T}\left(t\right)e_{i}\left(t\right){)}^{\frac{\beta +1}{2}}+\kappa_{3}\sum_{i=1}^{N}e_{i}^{T}\left(t\right)e_{i}\left(t\right)+\frac{N}{\kappa_{3}}W^{T}W\\ \leq & -2e^{T}\left(t\right)(I_{N}⊗C_{1})e\left(t\right)+2e^{T}\left(t\right)(I_{N}⊗\left|A_{1}\right|LI)e\left(t\right)+\kappa_{1}e^{T}\left(t\right)e\left(t\right)\\ \; & +2e^{T}\left(t\right){\tilde{D}}_{1}e\left(t\right)+2\kappa_{2}e^{T}\left(t\right)e\left(t\right)-2\lambda e^{T}\left(t\right)e\left(t\right)-2\sigma_{1}(e^{T}\left(t\right)e\left(t\right){)}^{\frac{\alpha +1}{2}}\\ \; & -2\sigma_{2}{\left(2nN\right)}^{\frac{1-\beta }{2}}{\left(e^{T}\left(t\right)e\left(t\right)\right)}^{\frac{\beta +1}{2}}+\frac{N}{\kappa_{1}}\left(\right|A_{1}|h{)}^{T}\left|A_{1}\right|h\\ \; & +\frac{2N}{\kappa_{2}}\left(\right|B_{1}|M{)}^{T}\left|B_{1}\right|M+\kappa_{3}e^{T}\left(t\right)e\left(t\right)+\frac{N}{\kappa_{3}}W^{T}W\\ \leq & \left(\lambda_{\max}\left(\Phi \right)+\kappa_{1}+2\kappa_{2}+\kappa_{3}-2\lambda \right)e^{T}\left(t\right)e\left(t\right)-2\sigma_{1}{\left(e^{T}\left(t\right)e\left(t\right)\right)}^{\frac{\alpha +1}{2}}\\ \; & -2\sigma_{2}{\left(2nN\right)}^{\frac{1-\beta }{2}}{\left(e^{T}\left(t\right)e\left(t\right)\right)}^{\frac{\beta +1}{2}}+\frac{N}{\kappa_{1}}\left(\right|A_{1}|h{)}^{T}\left|A_{1}\right|h\\ \; & +\frac{2N}{\kappa_{2}}\left(\right|B_{1}|M{)}^{T}\left|B_{1}\right|M+\frac{N}{\kappa_{3}}W^{T}W, \end{array} \end{array}$$
where $\Phi =-I_{N}⊗\left(C_{1}+C_{1}^{T}\right)+I_{N}⊗\left(\right|A_{1}\left|LI+(\right|A_{1}{\left|LI\right)}^{T})+\left({\tilde{D}}_{1}+{\tilde{D}}_{1}^{T}\right)$.Therefore,
$$\begin{array}{cc} \dot{V}\left(t\right)\leq & -\lambda_{1}V\left(t\right)-2\sigma_{1}V{\left(t\right)}^{\frac{\alpha +1}{2}}-2\sigma_{2}{\left(2nN\right)}^{\frac{1-\beta }{2}}V{\left(t\right)}^{\frac{\beta +1}{2}}+ℵ \end{array}$$
for $t\in [\zeta_{k},\mu_{k})$, $\lambda_{1}=2\lambda -\lambda_{\max}\left(\Phi \right)-\kappa$, $ℵ=\frac{N}{\kappa_{1}}\left(\right|A_{1}|h{)}^{T}|A_{1}|h+\frac{2N}{\kappa_{2}}\left(\right|B_{1}|M{)}^{T}\left|B_{1}\right|M+\frac{N}{\kappa_{3}}W^{T}W$, and $\kappa =\kappa_{1}+2\kappa_{2}+\kappa_{3}$.Then, for $t\in [\mu_{k},\zeta_{k+1})$, we have
$$\begin{array}{cc} \dot{V}\left(t\right)\leq & 2e^{T}\left(t\right)[-I_{N}⊗C_{1}+I_{N}⊗\left|A_{1}\right|LI+{\tilde{D}}_{1}]e\left(t\right)+\left(\psi +\kappa \right)e^{T}\left(t\right)e\left(t\right)\\ \; & +\frac{N}{\kappa_{1}}\left(\right|A_{1}|h{)}^{T}|A_{1}|h+\frac{2N}{\kappa_{2}}\left(\right|B_{1}|M{)}^{T}\left|B_{1}\right|M\\ \leq & \tilde{\psi }V\left(t\right)+ℵ, \end{array}$$
where $\tilde{\psi }=\lambda_{\max}\left(\Phi \right)+\psi +\kappa$.Based on Lemma 3, the networks (12) and (13) achieve FXT quasi-bipartite synchronization and the settling time is estimated as $T^{*}$. Moreover, the system error e(t) will converge to $\Omega =\left\{e\left(t\right)|{∥e\left(t\right)∥}_{2}\leq \sqrt{\max\{\delta_{1},\delta_{2},\delta_{3}\}}\right\}$ within *T*, where $\delta_{1}={\left(\frac{ℵ}{2\phi \sigma_{2}{\left(nN\right)}^{\frac{1-\beta }{2}}}\right)}^{\frac{2}{\beta +1}}$, $\delta_{2}={\left(\frac{ℵ}{2\phi \sigma_{1}}\right)}^{\frac{2}{\alpha +1}}$, $\delta_{3}=\frac{ℵ}{d\gamma -\tilde{\psi }}$.The theorem is proven. □

**Remark** **5.**
*A previous study [23] focused on studying finite-time bipartite synchronization in competitive neural networks, whereby the settling time was dependent on the initial state. In contrast, Theorem 1 provides a sufficient condition for achieving FXT quasi-bipartite synchronization in CCNNs where the settling time is no longer dependent on the initial state, but rather on the adjustable controller parameters and the average control rate. Furthermore, our study utilizes a more practical intermittent controller compared to the one used in [23].*


In particular, when the external disturbance of the network is zero, the following Corollary 1 can be obtained. Obviously, it can be seen that the convergence domain of the network is smaller and closer to complete synchronization without external disturbance.

**Corollary** **1.**
*Based on Assumptions 1–5 and the controller (8), if*

$$\begin{array}{cc} \; & 2\lambda -\lambda_{\max}\left(\Phi \right)-\tilde{\kappa }>0,\\ \; & \varrho_{1}+\lambda_{\max}\left(\Phi \right)+\tilde{\kappa }>0,\\ \; & \varrho_{1}+\lambda_{\max}\left(\Phi \right)+\tilde{\kappa }-\gamma d<0,\\ \; & \left(1-\gamma \right)d-2\lambda +\lambda_{\max}\left(\Phi \right)+\tilde{\kappa }<0, \end{array}$$

*where $\Phi =-I_{N}⊗\left(C_{1}+C_{1}^{T}\right)+I_{N}⊗\left(|A_{1}\left|LI+(\right|A_{1}{\left|LI\right)}^{T}\right)+({\tilde{D}}_{1}+{\tilde{D}}_{1}^{T})$, $\tilde{\kappa }=\kappa_{1}+2\kappa_{2}$, λ, $\varrho_{1}$, d are positive constants, γ∈(0,1) stands for the average control rate, then the quasi-bipartite synchronization can be ensured between networks (12) and (13) in fixed time. The settling time satisfies*

T≤2+a5β−1Tγa5β−1γ+2+a61−αTγa61−αγ,

*here $a_{5}=2\sigma_{2}{\left(2nN\right)}^{\frac{1-\beta }{2}}\left(1-\phi \right)\exp\left\{\frac{1}{2}T_{\gamma }\left(1-\beta \right)d\right\}$, $a_{6}=2\sigma_{1}\left(1-\phi \right)$, $T_{\gamma }$ represents the elasticity number, 0<ϕ<1. The state trajectory of (14) converges to a compact set $\tilde{\Omega }=\left\{e\left(t\right)|{∥e\left(t\right)∥}_{2}\leq \sqrt{\max\{{\tilde{\delta }}_{1},{\tilde{\delta }}_{2},{\tilde{\delta }}_{3}\}}\right\}$, ${\tilde{\delta }}_{1}={\left(\frac{\tilde{ℵ}}{2\phi \sigma_{2}{\left(nN\right)}^{\frac{1-\beta }{2}}}\right)}^{\frac{2}{\beta +1}}$, ${\tilde{\delta }}_{2}={\left(\frac{\tilde{ℵ}}{2\phi \sigma_{1}}\right)}^{\frac{2}{\alpha +1}}$, ${\tilde{\delta }}_{3}=\frac{\tilde{ℵ}}{d\gamma -\varrho }$, $\varrho =\varrho_{1}+\lambda_{\max}\left(\Phi \right)+\tilde{\kappa }$, $\tilde{ℵ}=\frac{N}{\kappa_{1}}{\left(|A_{1}\left|h\right)\right)}^{T}|A_{1}|h+\frac{2N}{\kappa_{2}}{\left(|B_{1}|M\right)}^{T}\left|B_{1}\right|M$.*


**Proof.** The proof process of Corollary 1 is the same as Theorem 1, and it is not repeated. □

**Remark** **6.**
*Unlike prior studies [34,37], our research takes into account the impact of discontinuous activation functions, external disturbances, and competitive relationships between nodes, which more closely mimics real-world networks. Specifically, due to the competitive nature among nodes and the presence of external disturbances, the synchronization method employed in [34] is not directly applicable to achieve FXT quasi-bipartite synchronization. Consequently, our main theorem extends the prior findings of FXT bipartite synchronization and is tailored to suit the aforementioned conditions.*


If the time-varying delay, discontinuous activation functions, and external disturbance are not considered, Corollary 2 is obtained. The error of Corollary 2 will converge to 0, and the result is fixed-time complete bipartite synchronization.

**Corollary** **2.**
*Based on Assumptions 1–4 and the controller (8), if*

$$\begin{array}{cc} \; & 2\lambda -\lambda_{\max}\left(\Phi \right)>0,\\ \; & \varrho_{1}+\lambda_{\max}\left(\Phi \right)>0,\\ \; & \varrho_{1}+\lambda_{\max}\left(\Phi \right)-\gamma d<0,\\ \; & \left(1-\gamma \right)d-2\lambda +\lambda_{\max}\left(\Phi \right)<0, \end{array}$$

*where $\Phi =-I_{N}⊗\left(C_{1}+C_{1}^{T}\right)+I_{N}⊗\left(|A_{1}\left|LI+(\right|A_{1}{\left|LI\right)}^{T}\right)+({\tilde{D}}_{1}+{\tilde{D}}_{1}^{T})$, λ, $\varrho_{1}$, d are positive constants, γ∈(0,1) stands for the average control rate, then the bipartite synchronization can be ensured between networks (12) and (13) in fixed time. The settling time satisfies*

T≤2+a5β−1Tγa5β−1γ+2+a61−αTγa61−αγ,

*Here, $a_{5}=2\sigma_{2}{\left(2nN\right)}^{\frac{1-\beta }{2}}\left(1-\phi \right)\exp\left\{\frac{1}{2}T_{\gamma }\left(1-\beta \right)d\right\}$, $a_{6}=2\sigma_{1}\left(1-\phi \right)$, $T_{\gamma }$ represents the elasticity number, 0<ϕ<1.*


**Proof.** The proof process of Corollary 2 is also similar to the proof process of Theorem 1, and the proof is no longer repeated. □

**Remark** **7.**
*It can be seen from Corollary 2 that time-varying delay, discontinuous activation function, and external disturbance will make the network unable to achieve complete synchronization, and only quasi-synchronization can be achieved. However, the results of [38] show that competitive neural networks with time-varying delays and discontinuous activation functions can achieve fixed-time complete synchronization, rather than quasi-synchronization. After analysis, it can be found that due to the intermittent control strategy, the problem of achieving full synchronization of fixed time under intermittent control is worth studying in the future.*


**Remark** **8.**
*Predefined time control has emerged as a promising method that allows synchronization time to be pre-set independently of system parameters. Due to its potential in various applications, predefined time synchronization has become a highly topical research area. However, there is still insufficient research into the predefined time bipartite synchronization of CCNNs, therefore further investigation is necessary.*


## 4. Numerical Examples

Two numerical examples are given in this part to demonstrate the validity of the derived theoretical conclusions.

Consider the following network:(22)$$\left\{\begin{array}{c} \begin{array}{cc} \dot{z}\left(t\right)= & -\frac{1}{\varepsilon }Cz\left(t\right)+\frac{1}{\varepsilon }Af\left(z\left(t\right)\right)+\frac{1}{\varepsilon }Bf\left(z\left(t-\tau \left(t\right)\right)\right)+\frac{1}{\varepsilon }Es\left(t\right),\\ \dot{s}\left(t\right)= & -\bar{C}s\left(t\right)+\bar{A}f\left(z\left(t\right)\right), \end{array} \end{array}\right.$$
where $z\left(t\right)={\left(z_{1}\left(t\right),z_{2}\left(t\right)\right)}^{T}$, $s\left(t\right)={\left(s_{1}\left(t\right),s_{2}\left(t\right)\right)}^{T}$, $y\left(t\right)={\left(z^{T}\left(t\right),s^{T}\left(t\right)\right)}^{T}$, ε=0.83, $\tau \left(t\right)=\frac{2e^{t}}{2(1+e^{t})}$, $f\left(z\left(t\right)\right)={\left(\sin\left(z_{1}\left(t\right)\right)+0.01\mathrm{sign}\left(z_{1}\left(t\right)\right),\sin\left(z_{2}\left(t\right)\right)+0.01\mathrm{sign}\left(z_{2}\left(t\right)\right)\right)}^{T}$, C=diag(2,2), $\bar{C}=\mathrm{diag}\left(0.5,0.6\right)$, $\bar{A}=\mathrm{diag}\left(1.2,1\right)$, E=diag(0.01,0.01), A=[1,2;−5.2,3.2], B=[−4,2.1;2,3.5]. Figure 1 shows the chaotic trajectories of network (22) with initial values $z\left(0\right)={\left(-0.4,0.6\right)}^{T}$, $s\left(0\right)={\left(-0.7,0.1\right)}^{T}$.

The competitive neural network considering seven nodes coupling is as follows:(23)$$\left\{\begin{array}{c} \begin{array}{cc} {\dot{x}}_{i}\left(t\right)= & -\frac{1}{\varepsilon }Cx_{i}\left(t\right)+\frac{1}{\varepsilon }Af\left(x_{i}\left(t\right)\right)+\frac{1}{\varepsilon }Bf\left(x_{i}\left(t-\tau \left(t\right)\right)\right)+\frac{1}{\varepsilon }Ey_{i}\left(t\right)\\ \; & +\sum_{j=1}^{7}\left|d_{ij}\right|\left(\mathrm{sign}\left(d_{ij}\right)x_{j}\left(t\right)-x_{i}\left(t\right)\right)+R_{i}\left(t\right)+\Xi_{i},\\ {\dot{y}}_{i}\left(t\right)= & -\bar{C}y_{i}\left(t\right)+\bar{A}f\left(x_{i}\left(t\right)\right)+\sum_{j=1}^{7}\left|u_{ij}\right|\left(\mathrm{sign}\left(u_{ij}\right)y_{j}\left(t\right)-y_{i}\left(t\right)\right)\\ \; & +{\bar{R}}_{i}\left(t\right)+\Xi_{i},\;i=1,2,\cdots ,7, \end{array} \end{array}\right.$$
where $x_{i}\left(t\right)={\left(x_{i1}\left(t\right),x_{i2}\left(t\right)\right)}^{T}$, $y_{i}\left(t\right)={\left(y_{i1}\left(t\right),y_{i2}\left(t\right)\right)}^{T}$, $x_{i}\left(t\right)={\left(x_{i}^{T}\left(t\right),y_{i}^{T}\left(t\right)\right)}^{T}$, and where $\Xi_{i1}\left(t\right)=0.2\cos\left(t\right)$, $\Xi_{i2}\left(t\right)=0.1\sin\left(2t\right)$.

The topology of the network (23) is presented in Figure 2. Let $V_{1}=\left\{1,2,3\right\}$, $V_{2}=\left\{4,5,6,7\right\}$, and take ω=diag(1,1,1,−1,−1,−1,−1), ${\left(e_{i}^{T}\left(t\right),{\hat{e}}^{T}\left(t\right)\right)}^{T}=x_{i}\left(t\right)-w_{i}y\left(t\right)$. It satisfies the definition of structurally balanced for a signed graph.

The control intervals of the intermittent controller (8) is designed as follows:$$\overset{+\infty }{\underset{k=0}{⋃}}[\zeta_{k},\mu_{k})=\overset{+\infty }{\underset{l=0}{⋃}}[0.2l,0.2l+0.05)\cup \left[0.2l+0.1,0.2l+0.18\right).$$

Set the initial values of network (23) to be $x_{1}\left(v\right)={\left(-1.9,2.8\right)}^{T}$, $x_{2}\left(v\right)={\left(-1.3,-1.4\right)}^{T}$, $x_{3}\left(v\right)={\left(2.2,1.4\right)}^{T}$, $x_{4}\left(v\right)={\left(-2.1,-2.9\right)}^{T}$, $x_{5}\left(v\right)={\left(2.4,-1.8\right)}^{T}$, $x_{6}\left(v\right)={\left(-1.2,0.9\right)}^{T}$, $x_{7}\left(v\right)={\left(-1.3,0.2\right)}^{T}$, $y_{1}\left(v\right)={\left(-2.6,0.1\right)}^{T}$, $y_{2}\left(v\right)={\left(0.4,-0.1\right)}^{T}$, $y_{3}\left(v\right)={\left(0.1,-0.9\right)}^{T}$, $y_{4}\left(v\right)={\left(-0.4,-1.6\right)}^{T}$, $y_{5}\left(v\right)={\left(0.5,1.6\right)}^{T}$, $y_{6}\left(v\right)={\left(0.1,0.8\right)}^{T}$, $y_{7}\left(v\right)={\left(-1.7,0.7\right)}^{T}$, v∈[−1,0]. By simple calculation, γ=0.65 and λ=32. Choose α=0.96, β=3, $\sigma_{1}=20$, $\sigma_{2}=25$, $\kappa_{1}=0.01$, $\kappa_{2}=10$, ϕ=0.5, $\kappa_{3}=1$, d=101 and $T_{\gamma }=0.0001$. By Theorem 1, the networks (22) and (23) are FXT quasi-bipartite synchronization, and it is obtained that T=5.59 s and error bound is 2.97.

Under the controller (8), Figure 3a,b depict the trajectories of the first and second components of STM, and Figure 3c,d depict the trajectories of the first and second components of LTM. The time evolution of synchronization errors $e_{i}\left(t\right)=x_{i}\left(t\right)-w_{i}z\left(t\right)$ and ${\hat{e}}_{i}\left(t\right)=y_{i}\left(t\right)-w_{i}s\left(t\right)$ between systems (22) and (23) are depicted in Figure 4. Define $E_{1}\left(t\right)={∥e\left(t\right)∥}_{2}^{2}$ and $E_{2}\left(t\right)={∥\hat{e}\left(t\right)∥}_{2}^{2}$, in which $e\left(t\right)={\left(e_{1}\left(t\right),\cdots ,e_{7}\left(t\right)\right)}^{T}$ and $\hat{e}\left(t\right)={\left({\hat{e}}_{1}\left(t\right),\cdots ,{\hat{e}}_{7}\left(t\right)\right)}^{T}$. Figure 5a,b show that the synchronization errors eventually converge within a bounded region. Together with Figure 3 and Figure 4, it can be seen that systems (22) and (23) under the intermittent controller (8) achieve the quasi-bipartite synchronization in fixed time, which coincides with the conclusion of Theorem 1.

## 5. Conclusions

In this study, the problem of FXT quasi-bipartite synchronization for CCNNs is considered by intermittent control strategy. Compared with the existing FXT intermittent control strategy, the linear term on the rest interval is removed, which makes our control method simpler and more economical. In addition, the influence of discontinuous activation functions, external disturbances, and competitive relationships between nodes are considered in synchronous analysis, which makes the obtained criteria more general. Note that we can implement control measures on all nodes to achieve FXT synchronization when the network topology is known. In practical scenarios, it is often either infeasible or unnecessary to control every single node. Therefore, the intermittent pinning control will be considered in forthcoming research.

## Figures and Tables

**Figure 1 entropy-26-00199-f001:**
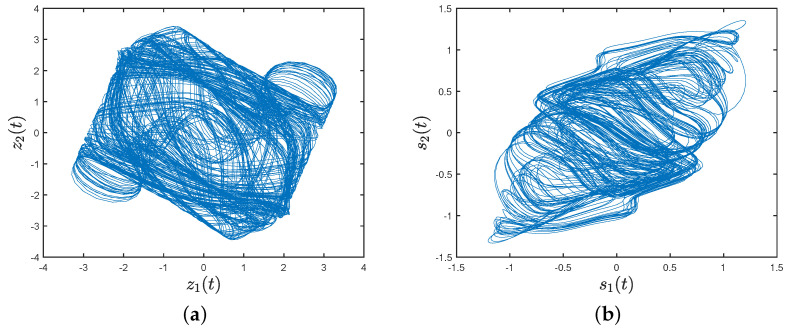
(**a**) Chaotic trajectory of z(t); (**b**) chaotic trajectory of s(t).

**Figure 2 entropy-26-00199-f002:**
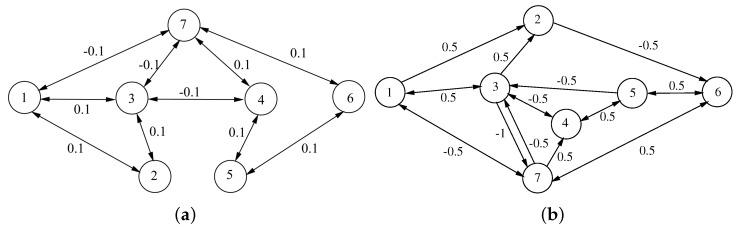
(**a**) Topology structure of STM in network (23); (**b**) topology structure of LTM in network (23).

**Figure 3 entropy-26-00199-f003:**
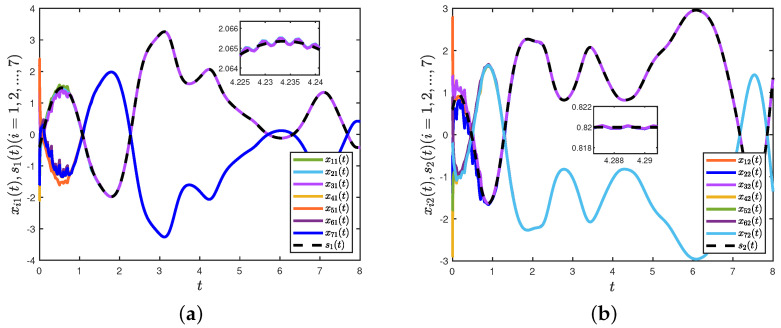
(**a**) Trajectories of the first component of $x_{i}\left(t\right)$; (**b**) trajectories of the second component of $x_{i}\left(t\right)$; (**c**) trajectories of the first component of $y_{i}\left(t\right)$; (**d**) trajectories of the second component of $y_{i}\left(t\right)$.

**Figure 4 entropy-26-00199-f004:**
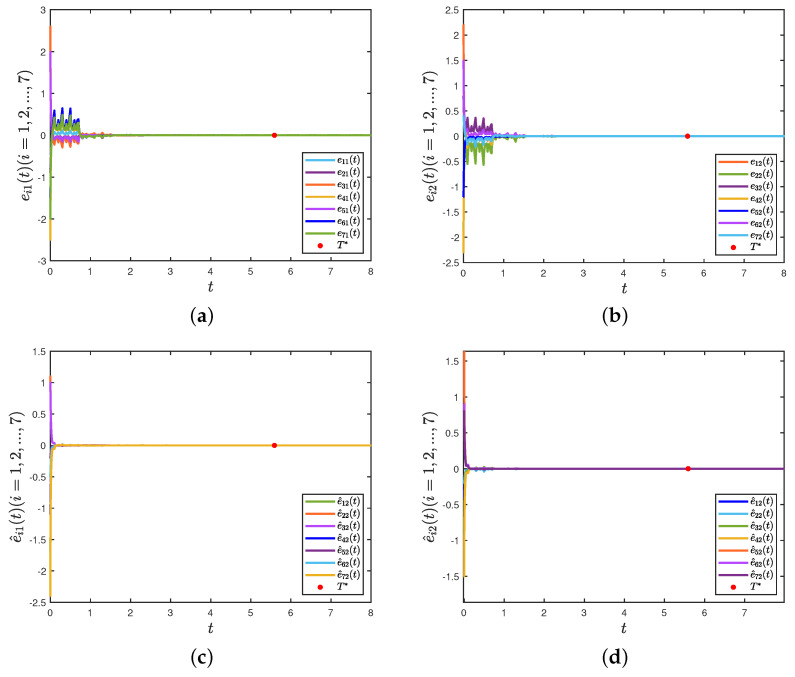
(**a**) Trajectories of the first component of $e_{i}\left(t\right)$; (**b**) trajectories of the second component of $e_{i}\left(t\right)$; (**c**) trajectories of the first component of ${\hat{e}}_{i}\left(t\right)$; (**d**) trajectories of the second component of ${\hat{e}}_{i}\left(t\right)$.

**Figure 5 entropy-26-00199-f005:**
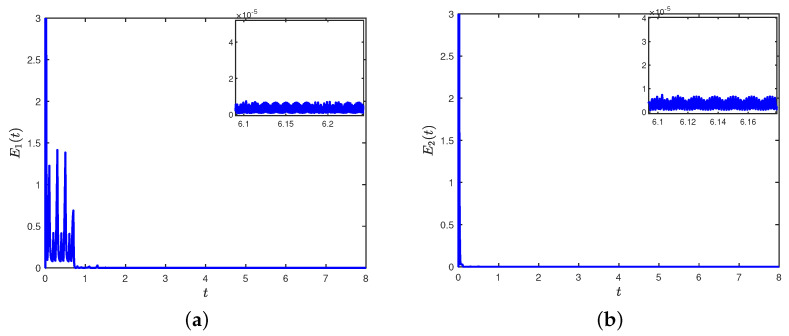
(**a**) Trajectory of $E_{1}\left(t\right)$ with $E_{1}\left(t\right)={∥e\left(t\right)∥}_{2}^{2}$; (**b**) trajectory of $E_{2}\left(t\right)$ with $E_{2}\left(t\right)={∥\hat{e}\left(t\right)∥}_{2}^{2}$.

## Data Availability

The data presented in this study are available on request from the corresponding author.

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
