# Peer review of "Fixed-Time Aperiodic Intermittent Control for Quasi-Bipartite Synchronization of Competitive Neural Networks"

_entropy, 2024, doi:10.3390/e26030199_

Round 1

Reviewer 1 Report

Comments and Suggestions for Authors

In this paper, the fixed-time quasi-bipartite synchronization of competitive neural networks is investigated by means of aperiodic intermittent control. The obtained results seem to be correct. However, some comments need be considered:

(1) Theorem 1 and Theorem 2 are almost identical. What is the difference between them? Can Theorem 1 be deemed as the corollary of Theorem 2?

(2) In Introduction, the main contributions of this paper should be summarized.

(3) In recent years, there have been many results about the fixed-time synchronization and predefined-time synchronization of all kinds of neural networks. Compared to these achievements, what is the novelty of this paper?

(4) What are the main difficulties in this study?

Comments on the Quality of English Language

The English writing should be polished.

Author Response

\reviewersection

\begin{point}
Theorem 1 and Theorem 2 are almost identical. What is the difference between them? Can Theorem 1 be deemed as the corollary of Theorem 2?
\end{point}

\begin{reply}
{
We sincerely thank you for your invaluable suggestion. 
The distinction between Theorem 1 and Theorem 2 lies in the presence of an external disturbance. Theorem 1 can be considered a corollary of Theorem 2. To prevent redundancy in the content, we have implemented modifications in the article, utilizing red for clarity.
}
\end{reply}

\begin{point}
In Introduction, the main contributions of this paper should be summarized.
\end{point}

\begin{reply}
{
We appreciate your suggestion and acknowledge the importance of clearly outlining the key contributions of our work. We have enhanced the introduction section to provide a concise summary of the significant contributions outlined in red font in the revised version.
}
\end{reply}

\begin{point}
 In recent years, there have been many results about the fixed-time synchronization and predefined-time synchronization of all kinds of neural networks. Compared to these achievements, what is the novelty of this paper?

\end{point}

\begin{reply}
{
We sincerely appreciate your insightful inquiry. Firstly, our paper addresses competitive neural networks, which encompass two state variables, adding a layer of realism not commonly found in general neural network studies. Secondly, there has been limited exploration of coupled competitive neural networks in recent research. Notably, we focus on achieving fixed-time quasi-bipartite synchronization within such networks, considering both cooperation and competition among nodes within a defined timeframe. Additionally, we employ an intermittent control strategy, recognized for its cost-effectiveness and practical applicability due to its discontinuous nature. Lastly, our work distinguishes itself by considering practical challenges such as time delays and the use of discontinuous activation functions, factors often encountered in real-world applications.

}
\end{reply}

\begin{point}
What are the main difficulties in this study?

\end{point}

\begin{reply}
{
We sincerely appreciate your valuable feedback. The main difficulties encountered in this study revolve around addressing time delays, discontinuous activation functions, and external disturbances within the framework of an intermittent control strategy. The uncontrolled rest periods inherent to intermittent control exacerbate the complexities associated with time delays, discontinuous activation functions, and external disturbances. Furthermore, our investigation into achieving fixed-time bipartite synchronization is compounded by the compression of the two-layer structure of coupled competitive neural networks into a single-layer high-dimensional structure.
}
\end{reply}

Reviewer 2 Report

Comments and Suggestions for Authors

In this manuscript the authors present a mathematical study on the control of neural networks (NN).

Actually, the present a nice motivation, then proceed to formal results in the form of definitions, theorems and consequences, and finally they present some illustrative results.

Moreover, it is very well structured in sections (I like in this respect that they make a section with the Main result), and well written, despite grammatical mistakes

(see comments below).

As such, the manuscript is interesting and, although the results are not earth-shaking, I may find that it deserves publication in Entropy.

However, I will raise some comments that, if properly taken into account, they will undoubtedly contribute to improve the final result.

In particular:

1) In line 82 change 'R' to the proper symbol of real number set, which in Latex is: $\mathbb{R}$, provided that the suitable packages are used, i.e. \usepackage{amsmath} and \usepackage{amssymb}.

2) At the beginning of Sect 2, the authors state 'Consider the following competitive neural network.... and jump to write down a system of ODE in Eq. (1). I think that here it is mandatory to make a subsection in which the relation of NN and ODE's is explicitly explained. This will make the paper suitable for a broader audience.

3) Figures 4, 5, 7, and 8 are very poorly designed, and the different graphs are hardly seen. Please, fix.

4) Discussion and interpretation of the results in subsect 4.1 and 4.2 are missing. This is not acceptable. Also discuss somewhere the importance of the reported results.

For example, at the end of page 14 described what Fig. 2 means, and its relevance.

5) In Fig. 1 there is only 1 trajectory per graph? (then eliminate the plural) or there are many? then how many? (then specify the number). How were those trajectories selected?

5) Grammatical mistakes (some):

- line 13 'only' is strange here.

- line 20. Include space between 'networks' and '(CCNNs)'. Is not a 'C' superfluous here?

- In line 28 insert 'e' in changes'

- In line 29 change 'aware' to 'aware of' and 'studies take' to 'studies that take'

- In paragraph 46-56 too many time the word 'synchronization' appears, making it cacophonous.

- In line 78 why not starting the organization of the paper as it is customary?, i.e. The organization of this paper is as follows.

- In paragraph 95-100 too many sentences start with a math symbol. Change that to comma separations.

Also, the final sentence is started by 'where' that in my opinion is incorrect: 'where' should only be used after formulae.

- In line 102 change 'losing' to 'loss of'

Comments on the Quality of English Language

Needs a lot of revision

Author Response

\reviewersection

\begin{point}
In line 82 change 'R' to the proper symbol of real number set, which in Latex is: $\mathbb{R}$, provided that the suitable packages are used, i.e. $\setminus$usepackage\{amsmath\} and $\setminus$usepackage\{amssymb\}.
\end{point}

\begin{reply}
{
We gratefully appreciate for your valuable suggestion.
It has been modified in the article and marked in red.
}
\end{reply}

\begin{point}
At the beginning of Sect 2, the authors state 'Consider the following competitive neural network.... and jump to write down a system of ODE in Eq. (1). I think that here it is mandatory to make a subsection in which the relation of NN and ODE's is explicitly explained. This will make the paper suitable for a broader audience.
\end{point}

\begin{reply}
{ Thank you for your valuable suggestions.
Neural network models are often formulated in terms of Ordinary Differential Equations (ODEs), mainly because ODEs are a mathematical tool that effectively describes the dynamic behavior and interactions between neurons. Specifically:

~~First, the state of the neuron needs to be defined. This may include the neuron's potential, activity, or other relevant variables. These states will be the unknowns of the ODEs. 

~~Second, interaction rules need to be established. Rules describing the interactions between neurons are usually expressed in the form of weights and connections. These rules will determine how one neuron affects other neurons. Based on the states of neurons and the rules of interaction, a system of ODEs can be built to describe the dynamic evolution of a neural network. This usually involves differentiating the interaction rules to capture the temporal changes in the system. 
The ODEs system then needs to be provided with initial conditions, which represent the initial state of the neural network at a given moment in time. 

~~Finally, the system of ODEs can be solved using numerical methods (e.g., Euler's method, Runge-Kutta method, etc.) or analytical methods to obtain the state of the neural network at different points in time. 
In this way, the dynamic behavior of the neural network can be accurately represented in mathematical form. This model of ODEs not only helps in theoretical analysis, but also provides a basis for simulation and emulation, enabling us to better understand the behavior and response of neural networks. 
\\

To enhance the article's readability, we have incorporated Remark 1 into the updated version.
}
\end{reply}

\begin{point}
Figures 4, 5, 7, and 8 are very poorly designed, and the different graphs are hardly seen. Please, fix.
\end{point}

\begin{reply}
{
We sincerely thank you for your valuable suggestions. 
Since Theorem 1 has been corrected to a corollary of Theorem 2, only numerical simulations in the presence of external disturbances are given to justify the conclusions.
}
\end{reply}

\begin{point}
Discussion and interpretation of the results in subsect 4.1 and 4.2 are missing. This is not acceptable. Also discuss somewhere the importance of the reported results.
For example, at the end of page 14 described what Fig. 2 means, and its relevance.
\end{point}

\begin{reply}
{
Thank you for your valuable suggestions.
We have added appropriate discussions and explanations in the revised version and highlighted them in red.  
}
\end{reply}

\begin{point}
In Fig. 1 there is only 1 trajectory per graph? (then eliminate the plural) or there are many? then how many? (then specify the number). How were those trajectories selected?
\end{point}

\begin{reply}
{
Thank you for your valuable suggestions. In Figure 1, there is only one  trajectory per graph. The plural form has been removed and is highlighted in red. 
}
\end{reply}

\begin{point}
Grammatical mistakes (some):

- line 13 'only' is strange here.

- line 20. Include space between 'networks' and '(CCNNs)'. Is not a 'C' superfluous here?

- In line 28 insert 'e' in changes'

- In line 29 change 'aware' to 'aware of' and 'studies take' to 'studies that take'

- In paragraph 46-56 too many time the word 'synchronization' appears, making it cacophonous.

- In line 78 why not starting the organization of the paper as it is customary?, i.e. The organization of this paper is as follows.

- In paragraph 95-100 too many sentences start with a math symbol. Change that to comma separations.

Also, the final sentence is started by 'where' that in my opinion is incorrect: 'where' should only be used after formulae.

- In line 102 change 'losing' to 'loss of'
\end{point}

\begin{reply}
{
We gratefully appreciate for your valuable suggestion.
The corresponding grammatical mistakes have been modified and marked in red in the article.
}
\end{reply}
~~

~~We hope this update meets your expectations. If you have any further suggestions or require additional clarification, we would be more than willing to address them. Once again, we appreciate your time and thoughtful review.

Round 2

Reviewer 1 Report

Comments and Suggestions for Authors

The manuscript can be accepted now.

Reviewer 2 Report

Comments and Suggestions for Authors

In my opinion, the authors have adequately taken into accounto the points raised by the two referees, which resulted in an improved manuscript.

Accordingly, I can recommend its publication in its present form